# Exploring factors affecting the adoption and continuance usage of drone in healthcare: The role of the environment

**John Serbe Marfo**[1]*, **Kwadwo Kyeremeh**[2], **Pasty Asamoah**[1], **Matilda Kokui Owusu-Bio**[1], **Afia Frimpomaa Asare Marfo**[3]

**1** Supply Chain and Information Systems Department, Kwame Nkrumah University of Science and Technology, Kumasi, Ghana, **2** Department of Accountancy, Sunyani Technical University, Sunyani, Ghana, **3** Department of Pharmacy Practice, Kwame Nkrumah University of Science and Technology, Kumasi, Ghana

* serbemarfo@gmail.com

## Abstract

Drone technologies and healthcare delivery have attracted scholarly attention over the years. Studies have acknowledged the positive impact of the adoption and usage of drone technologies for healthcare delivery. We argue however that, knowledge is lacking on the role of the environment in drone technologies adoption, usage and continuance usage. An examination of 330 health facilities that engage in the use of drone services from Zipline Ghana showed that the environment inversely moderates the relationship between actual usage and intention to continue usage, suggesting that reducing the influence of environmental factors will increase the impact actual usage has on the continuance usage of drone technology in healthcare delivery.

## Author summary

Our comprehensive study delves into the intriguing intersection of drone technology and healthcare delivery, an area of significant scholarly interest. Our research addresses a critical knowledge gap by investigating how environmental factors influence the adoption and ongoing use of drone technology in healthcare. Through an extensive analysis involving 330 healthcare facilities, with Zipline Ghana as a prominent example, we reveal an insightful relationship. We contend that the environment plays a pivotal role in moderating the link between actual drone usage and the intention to continue using this technology for healthcare purposes. Our findings indicate that reducing the impact of environmental factors can substantially increase the influence of actual usage on sustained drone technology utilization in healthcare delivery. Our research not only contributes to the ongoing discourse on drones in healthcare but also emphasizes the importance of considering environmental variables in shaping the adoption and continued utilization of such technology. We anticipate that our findings will inform policymakers, healthcare professionals, and stakeholders, promoting the more effective integration of drones to enhance healthcare delivery, benefiting both scientific and non-scientific audiences.

**Data Availability Statement:** We have published the dataset and supporting documents on figshare. The associated data is the participants response to the items measurements, demographic data,

SmartPLS results, SmartPLS bootstrap results, and the structural model. The corresponding DOI is https://doi.org/10.6084/m9.figshare.24188283.v1.

**Funding:** The authors received no specific funding for this work.

**Competing interests:** The authors have declared that no competing interests exist.

## 1. Introduction

Drone technologies and healthcare delivery have attracted scholarly attention over the years [1–7] The surge in the interest of drone technologies as an emerging technology especially in healthcare delivery can be attributed to the challenge of poor road and transportation systems, healthcare emergencies and the successes of drone deployment on large scale for health supply chains [8]. Drone technologies enhance emergency health delivery thereby saving lives [9,10]. An example is the use of drones in Ghana by Zipline, the largest medical drone technology company, to respond to the novel COVID-19 pandemic by transporting health supplies to health facilities in Ghana and transporting suspected COVID-19 patient blood samples from rural areas. While drone technology deployment in healthcare on large scale has been successful in developing countries such as Ghana and Rwanda [9], the same cannot be said of developed countries which are struggling to deploy drones for healthcare on large scale. Top news agencies like the Times Magazine, CNBC, and Tech Crunch in the United States have reported this phenomenon. In their articles, they indicate that the United States, as a developed country, had to hone its first medical drone delivery from countries such as Ghana and Rwanda [11]. This phenomenon where new ideas and solutions are accepted and evaluated in low and middle-income countries (e.g., Ghana) before spreading to high-income nations (e.g., United States of America) is known as reverse innovation [12]. It is important to understand the factors that are driving the adoption and continuance usage of drone technology which is leading to reverse innovation in the use of drones for healthcare.

Though, drone technologies and healthcare delivery literature acknowledge the positive impact of the adoption and usage of drone technologies for healthcare delivery [1,2,9,10], knowledge is limited on the factors accounting for the adoption and continuance usage of drones. Specifically, knowledge is lacking on the role of the environment in drone technologies adoption, usage and continuance usage. It is important to address these knowledge gaps because most developing country citizens in Africa rely on public health facilities knocked by the challenges of chronic drug and health supplies shortage [13]. However, the use of drone technologies enhances emergency health delivery by shuttling health supplies to these public health facilities within a minimum period of time [9] which can lead to saving the lives of patients. Examining the factors that influence drone technologies adoption will encourage its adoption to enhance health delivery performance. In addition, the area of drone technologies usage in health care delivery especially on large-scale deployment is relatively underresearched [14]. In this study, we examine how the environment accounts for drone technologies adoption, usage and continuance usage in healthcare delivery.

Furthermore, it has been shown that there is a high degree of drone technologies adoption for healthcare delivery in developing countries like Rwanda, Botswana, Tanzania, Nigeria, and Ghana [15] than developed countries [16]. This provides a justification for conducting the study in a developing economy like Ghana to examine the factors contributing to the successes and continual usage of drone technologies in the healthcare industry.

This study makes two (2) main contributions to the drone technologies and healthcare delivery literature. First, while studies have submitted that environmental factors affect drone technologies adoption and the relationship between usage and continuance usage [16,17], theoretical specification and empirical analysis on how the environment affects the relationship is lacking. The study fills this gap by merging the expectation and confirmation theory (ECT) and the technology organization environment theory (TOE) to generate insight on how the environment influences the adoption, actual usage and intentions to continuance usage. Finally, the study contributes to the reverse innovation theory by explaining how environmental factors (e.g., regulatory barriers) in developing countries can be leveraged to develop

capabilities for drone adoption in healthcare delivery among developed countries. We draw on the ECT to argue that user satisfaction with drone technologies in healthcare delivery drives its usage leading to their intention to continuous use. Leveraging the TOE, we argue that the adoption and continuous usage of drone technologies are influenced by the environment and that there is an indication of reverse innovation where the environment contributes immensely to drone technologies applications in healthcare delivery in developing countries.

## 2. Literature review

### 2.1. Drone technology use in healthcare

Drone technologies have been defined in the literature in diverse ways according to the context [4–6,18]. In the area of agriculture, drone technologies are essential unmanned aerial vehicles (UAV) used for observations, sensing, spraying pesticides and providing information to farmers [3,19]. In healthcare, drone technologies have been defined as UAV used for the transport of medical goods and services (e.g., emergency blood supplies, vaccines, medicines, diagnostic samples, and even organs) [6]. Common features identified among the various definitions submitted by authors are that drones are unpiloted, unmanned, aircrafts or aerial vehicles and remotely controlled [5,6,20,21].

The healthcare supply chain is faced with the challenge of poor distribution networks [22], unavailability of appropriate transport for logistics collection [8], and supply chain complexities [23]. This has resulted in health facilities adopting drone technologies for healthcare deliveries within and across borders [21,24,25]. A study conducted by [10] revealed that medical supplies in Ghana were hindered by the difficulty of transporting medical supplies from the central point to the point-of-care and other remote areas. In the quest to overcome this challenge, the Government of Ghana collaborated with a US-based company Zipline Technology to use drones in the supply of medical supplies for emergencies across the country. The report showed that the introduction of drone technologies for medical supplies in Ghana has significantly impacted positively on the emergency health delivery system. For example, the country of Ghana used drone technologies to respond to the novel COVID-19 pandemic by shuttling medical supplies to health facilities and suspected COVID-19 patients [9]. According to the authors, the deployment of drone technologies saved the lives of the general population.

### 2.2. The role of the environment in reverse innovation for drone technology use in healthcare

Reverse innovation (RI) has been defined in literature as new ideas and solutions that are accepted and evaluated in low and middle-income countries (e.g., Ghana) before spreading to high-income nations (e.g., Canada). RI has its foundational focus on business development and economics and was originally termed as "innovation blowback" in 2010 [12,26]. Studies have shown that financing, governance, health information systems, leadership, collaborations for novel diagnostics, medicines, vaccinations and health service delivery using drone technologies are all instances of effective reverse innovation [12,27,28]. Studies have shown that, reverse innovations is characterised by commodity performance that must be adapted to an acceptable level at a fraction of the cost of the adopted technology [29]; sustainability, which favours the deployment of technologies such as drone use in the delivery of healthcare [7]; removal or flexibility in legal and regulatory barriers that impede the implementation of new technologies and prevent rapid market access [17]; and meeting local geographical and environmental needs like the delivery of medicine to inaccessible populations during rainy seasons using drones [30].

The current economic disparity between developing markets and developed countries drives reverse innovation, which gives specialized answers to challenges that have not previously been handled in an affordable or culturally acceptable manner [28]. Unlike developed countries with strict regulatory barriers on the implementation of new technologies and aviation, developing countries have flexible policies and regulations that make it possible to implement drone technologies in the healthcare sector [16]. An instance of reverse innovation in a developing country is the deployment of drone technologies to transport health supplies with a large coverage in Ghana [10]. This was made possible because existing aviation laws and regulations on drones in Ghana are very flexible, making flying and delivery by drones very easy [9], compared to the advanced countries like the United States of America, which have strict laws and regulations on aviation. Such countries have to emulate strategies in the implementation of drone technologies in developing countries within the healthcare supply chain, making the theory of reverse innovation possible [16]. Experiences and lessons from reverse innovation in global health show that collaborations between developing countries and developed countries can result in system-wide advantages [30].

## 2.3. Actual and continuance usage of drones in healthcare

The actual and continuance usage of drone technologies explains the decision of users to either use and continue using drone technologies in healthcare delivery. Empirical evidences show that, the adoption and actual usage of drone technologies hinges on user satisfaction, performance [29], sustainability [7], flexibility in regulations and the implementation of new technologies [17,31], and meeting environmental needs [30]. This explains that the unavailability of such flexibilities and abilities in a country may impose difficulties in drone technologies adoption and usage which explains why developed countries are unable to easily implement drone technologies in the healthcare supply chain [9]. It can therefore be inferred that, the adoption and continual usage of drone technologies in the healthcare supply chain of developing countries like Ghana (transports health supplies) and Rwanda (transports blood) is as a result of the aforementioned characteristics. A study conducted by [32] in examining information systems continuance usage revealed that, actual usage of information systems or technologies plays a mediating role in determining continuance usage behaviour. This implies that, until the consumers actually consume or are involved in the drone experience either by themselves, families or relatives, they will still hold on to their prior usage expectations [33]. And this makes it necessary to examine the conditions for the adoptions and continual usage of drone technologies in healthcare delivery leveraging recent developments in developing countries (reverse innovation).

## 2.4. Theoretical framework

The study is grounded in the expectation and confirmation theory (ECT) and the technology organization environment theory (TOE). The ECT posits that satisfaction is directly influenced by disconfirmation of beliefs and perceived performance, and is indirectly influenced by both expectations and perceived performance by means of a mediational relationship which passes through the disconfirmation construct. Specifically, users compare their pre-purchased expectation and the post-purchase perceived performance as a means in determining satisfaction (confirmation) of a product or service, which will lead to repurchase intentions [30,34]. The theory has four (4) main constructs namely; expectations, performance, disconfirmation and satisfaction. Expectations serve as the anchor for comparison (anticipations), that is what clients will use to evaluate performance from a disconfirmation judgment [35,36]. Performance is users' evaluation after purchase or usage of a service (e.g., perception of product

quality). If a product meets or outperforms expectations (confirmation), post-purchase satis-
faction will result. If a product falls short of expectations (disconfirmation), the consumer is
likely to be dissatisfied [33]. ECT has a limitation in explaining the information system expec-
tation formation process [37]. For example, ECT primarily focuses on the individual level of
analysis and may overlook the social and contextual factors that influence expectations. In the
context of information systems, expectations are often shaped by factors such as social influ-
ence, cultural norms, organizational context, and technological advancements. It does not ade-
quately capture these external influences on expectation formation [38]. In addition, ECT is
centered around consumers repurchasing intentions of goods and services, handling of the
assumption and attitudes toward the goods and services attributes or performance [39]. It does
not capture the information systems quality factors (repair quality, information quality and
system quality) and subsequently does not explain the thought of end-user satisfaction of
information system frameworks [40]. We draw on the ECT to argue that users' satisfaction
with drone technologies in healthcare delivery drives its usage leading to their intention for
continuous use.

TOE is a theoretical framework that explains technology adoption in organizations and
describes how the process of adopting and implementing technological innovations are influ-
enced by the technological—all of the technologies that are relevant to the firm either in use or
available on the market [41]; organizational- the characteristics and resources of the firm and
the amount of slack resources [42]; and environmental contexts- structure of the industry, the
presence or absence of technology service providers, and the regulatory environment [3,31]. In
the area of healthcare, there is an indication of reverse innovation where the environmental
factors contribute immensely to the success story of its applications in developing countries
[43] and laws and regulations on aviation are not very strict. So, the continuance usage inten-
tions will be higher than those in the developed countries where the environmental factors are
not favorable for the use of drones within the healthcare supply chain [44]. We argue that the
adoption and continuous usage of drone technologies are influenced by the environment and
that there is an indication of reverse innovation where the environment contributes
immensely to drone technologies applications in healthcare delivery in developing countries.

## 2.5. Research model and hypothesis

### 2.5.1. Relationship between technology adoption, confirmation of expectations and
user satisfaction.   Empirical evidence shows that firm adoption of technologies is influenced
by their perception of usefulness, technical and organizational compatibility, complexity and
learning curve, pilot test/experimentation, and visibility/imagination [45,46]. We have pre-
sented the conceptual framework in Fig 1 and validated model in Fig 2. [47] asserts that cus-
tomer satisfaction is determined by the ease with which they can utilize the technological
service. The outcome may be that ease of use is a critical component of customer satisfaction
with technological services, as evidenced in the study of [47]. An easy-to-use system will
encourage higher system utilization, and users will be more inclined to embrace it if it is per-
ceived as such. [48] observed that, in order to sustain connectivity and communication
between smart city services (SCS), it is necessary to promote the adoption and usage of SCS
technology. Users must be able to see the benefits of accessing and completing SCS as an
improvement in their overall quality of lives [49]. The perceived benefits include the perceived
comfort, usefulness, convenience, and safety connected with the delivery of SCS [50].
Increased customer satisfaction is associated with lower price sensitivity and greater reputation
efficacy [33]. The complexity of smart services is linked to the discomfort experienced by users
who utilize SCS technology. Users' dissatisfaction is likely to have a negative impact on their

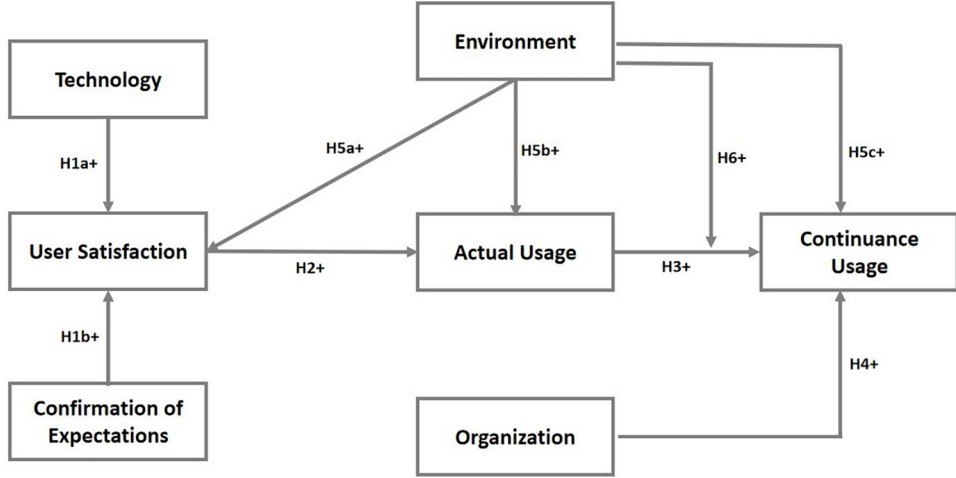

**Fig 1. Conceptual Framework.**

capacity to use the channels, as well as on their perceptions of SCS in general [51,52]. [53] opined that the experience of consumers with technology application is a major element in measuring consumer satisfaction with products and services.

The term "confirmation of expectations" was used by [54] in the confirmation of expectations theory to characterize the confirmation or disconfirmation of a person's pre-judgments (initial expectations) about a product or service [37]. According to [46], user satisfaction is determined by two elements: confirmation of expectations after actual usage and post-adoption expectations. Users' satisfaction may be measured against their expectations before they begin using an expectation or service [54]. Users' confirmation of expectations means that users received the expected value from their information technology (IT) usage experiences, which has a positive impact on users' satisfaction with their initial IT use [55]. Following the expectations confirmation model, the degree to which information system users confirm their expectations has a positive impact on their perceived usefulness and satisfaction with information systems (IS) [56]. After purchasing a product or service that meets or exceeds a person's

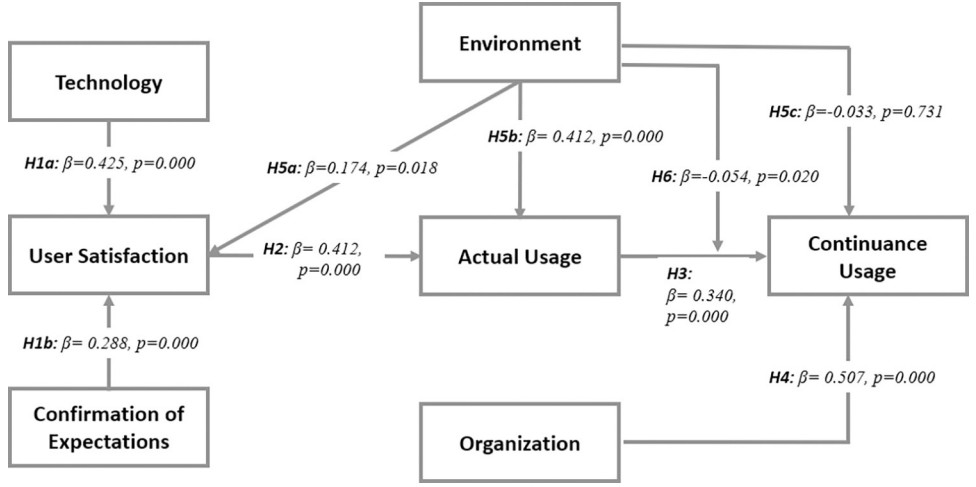

**Fig 2. Validated structural model.**

initial expectations, the confirmation is positive, resulting in an increase in product sales, which is referred to as post-adoption satisfaction. In every other case, the confirmation is negative, which lowers post-adoption satisfaction [57]. Following the arguments put forward on the relationship between technology adoption, confirmation of expectations and user satisfaction, the study hypothesize:

*H1a*: *There is a positive relationship between technology adoption and user satisfaction.*

*H1b*: *There is a positive relationship between confirmation of expectation and user satisfaction.*

**2.5.2. Relationship between user satisfaction and actual usage, and continuance.**   User satisfaction is an important factor in the success of e-services adoption [58]. Previous studies have shown that there is a link between users' satisfaction and continuous usage of an information system [41,59]. [32] posits that prior usage history, measured in terms of frequency and comprehensiveness mediates the direct relationship between user satisfaction and continuance usage of an information system. The study further stated that user satisfaction with an information system reinforces intentions to continue using the system. According to the findings of [60], satisfaction is an intrinsic motivation in the use of information systems. Perceived playfulness might be included as an additional intrinsic motive to the list of motivations typically identified in the research on information security adoption [61]. Customers' intentions to continue using a service are related with a service provider's capacity to gain and maintain client loyalty via guaranteeing customer satisfaction [62]. According to the expectation confirmation model in information systems, user satisfaction is a critical element in determining whether users want to continue using the system. Users' intention to remain a member of a brand community will be influenced by whether or not the brand community can give them satisfying perceived benefits [63]. Following these arguments, we posit that:

*H2*: *There is a direct relationship between user satisfaction and actual usage.*

*H3*: *There is a positive relationship between actual usage and continuance usage.*

**2.5.3. Organizational influence on continual usage.**   The continual usage of an information system is affected by the organizational context [64]. Studies have shown that the availability and characteristics of firm resources influences the continual usage of an information system [42]. [65] asserts that when implementing large and complex information systems and technology, critical consideration should be given, particularly with regard to top management support and facilitating conditions for the development of technology within medical environments. According to the authors organization captures descriptive measures such as the firm's business scope, top management support, organizational culture, the complexity of managerial structure as measured by centralization, formalization, and vertical differentiation, the quality of human capital, and size-related issues such as internal slack resources and specializations. In this study, organizational variables are addressed under the headings of just-in-time delivery, creativity, efficiency, innovations, and leadership [51]. Following the findings from literature, we hypothesis that:

*H4*: *There is a direct relationship between organization and continuance usage.*

**2.5.4. The role of the environment.**   Environmental factors have an impact on both operational facilitators and inhibitors; among the most significant are competitive pressure, the readiness of trading partners, sociocultural issues, government encouragement, and

technology support infrastructures such as access to qualified ICT consultants. In accordance with the findings of [66], environmental issues are a reflection of the current operating environment of the healthcare industry, and it is important to note that government policies should be taken into consideration when implementing new information systems and technologies in medical environments [67]. In this study, we consider government regulation, trading partners and access to resources as the environmental variables. Based on this, we argue that the environment directly influences the implementation of new information systems as well as strengthens the usage of an existing information system. We hypothesize:

*H5a*: *There is a positive relationship between environment and user satisfaction*

*H5b*: *There is a positive relationship between environment and actual usage*

*H5c*: *There is a positive relationship between environment and continuous usage*

*H6*: *Environment moderates the relationship between actual usage and continuous usage*

## 3. Methodology

### 3.1. Measures

The study explores how the environment influences the adoption, usage and continuance usage of drone technologies in the healthcare industry. The study items were sourced from extant literature. A number of considerations were made in the generation of the scale items in this step, including the extensive review of related literature, availability of existing measurement scale items, and how well the chosen item suits the hypotheses outlined in the previous chapters. These scale measurement items were carefully scrutinized to eliminate inappropriate items or items that do not measure the measurable construct in the hypothesis correctly. The initial measuring items were emailed to health professionals, drone service providers and information systems academicians for review. In total, seven (7) measuring items were used to realize the study objectives–technology adoption (measured with perceived barriers, perceived usefulness, perceived benefits, complexities and compatibility), user satisfaction (measured with prior usage and information system expectation), confirmation of expectation (measured with user expectations, perceived information system usage and perceived ease of use), actual usage, continuance usage, environment (measured with government regulations, trade partners and access to resources) and organization (measured with just-in-time delivery, creativity, efficiency, innovation and leadership). To facilitate the use of statistical analyses, a seven-point Likert scale with 7 representing "strongly agree" and 1 representing "strongly disagree" was used to measure the indicators. Demographic level data on firm characteristics were also collected and examined.

### 3.2. Sample and data collection

**3.2.1. Population and sampling.** Population as defined by [68], is a set of elements (persons or objects) that possess some common characteristics defined by the sampling criteria established by the researcher. In this study, the population (about 2300 health facilities) constituted health facilities involved in the use of Zipline's drone services and Zipline Ghana. Zipline is a medical delivery company headquartered in South San Francisco. The company operates in Ghana, Rwanda and other parts of Africa [9]. Specifically, the company engages in the transport of health supplies (vaccines, blood, etc.) using drone technologies. At the moment, Zipline is the only company that uses drone technologies to transport health logistics to the various health facilities in Ghana, providing a justification for the use of Zipline Ghana as a case study.

A convenience sampling approach was adopted to sample 330 out of the total population following the mathematical formula presented by [69]. [70] posits that convenience sampling enables the researcher to use value judgment to select cases that will best enable an individual to answer research questions and to meet objectives. According to the formular, sample size (n) = (p(100-p) z^2)/e where 'n' is the required sample size, 'p' is the percentage occurrence of a state or condition (0.5), 'e' is the percentage maximum error required (0.05) and 'z' is the value corresponding to level of confidence (1.96).

**3.2.2. Development of data collection instrument.**   The development of the data collection instrument for the study went through three (3) stages. First, the research constructs and questions were sourced from extant literature. Second, the developed questionnaire was emailed to two (2) healthcare delivery and drone technology researchers for examinations. Their comments helped refine the questionnaire. Third, the refined questionnaire was then emailed to two (2) additional healthcare facilities who adopt drone technologies in their service delivery for further examination. Following their comments, the final questionnaire was developed for data collection.

**3.2.3. Data collection.**   The study collected data from health facilities that engage in the use of Zipline's drone services as well as Zipline Ghana. The participants of the study included only staff who make use of drugs and health commodities delivered through drones in the identified health facilities. Questionnaires were distributed to the participants to collect data to answer the research questions. The researchers explained the questionnaires to the respondents thoroughly after copies had been given to them. The purpose was to help the respondents understand the content of the questionnaires and to do away with ambiguities, suspicions and also to be able to provide their independent opinions on the questionnaire items given to them. In total, 330 questionnaires were distributed out of which 312 (a response rate of 94.55%) questionnaires or responses were retrieved for the analysis. The researchers conducted an a priori power analysis to test the adequacy of the sample size. Using a recommended alpha value (α) of 0.05 and an anticipated effect size (Cohen's d) of 0.5 and a desired statistical power level of 0.8, a minimum of 102 sample size is required. Hence the retrieved sample of 312 questionnaires used in this study was sufficient to achieve a valid result.

# 4. Analysis

## 4.1. Demographic results

The study revealed that 232 (74.40%) of the respondents were males, and 80 (25.60%) of the respondents were females indicating that drone technology services have been deployed mostly in male dominated health facilities. This could be because there are less women with technological capabilities in Ghana. Also, 34 (10.90%) of the respondents were 20 years or less, 55 (17.60%) of the respondents were between the ages of 21 and 30 years, 185 (59.30%) of the respondents were between 31 and 39 years, 11 (3.50%) of the respondents were between the ages of 41 and 50, 2 (0.60%) respondents were between the ages of 51 to 60, and 25 (8.00%) of the respondents were over 61 years of age. Regarding educational level, 62 (19.50%) of the respondents were high school graduates, 126 (40.40%) bachelor degree holders, 56 (17.90%) professional certificate holders, 15 (4.80%) master's degree holders and 11 (3.50%) doctorate degree holders. Most of the respondents serve in private health facilities 212 (67.9%) indicating that the deployment of drone technology services for health delivery in Ghana is mostly seen in the private sectors. Respondents from public health facilities stood at 100 (32.10%). With respect to individual participants personal experience with drone, we found that 240 (76.90%) of the respondents have not had a direct experience with drone deliveries while 72 (23.10%) have had experience with drone deliveries. This indicates that while health facilities leverage

**Table 1. Demographic statistics.**

| Variable | Categories | Frequency | Frequency (%) |
|---|---|---|---|
| Gender | Male | 232 | 74.40% |
|  | Female | 80 | 25.60% |
| Age (years) | 20 and below | 34 | 10.90% |
|  | 21–30 | 55 | 17.60% |
|  | 31 to 39 | 185 | 59.30% |
|  | 41 to 50 | 11 | 3.50% |
|  | 51 to 60 | 2 | 0.60% |
|  | 61 and above | 25 | 8.00% |
| Highest Educational Level | High School | 62 | 19.90% |
|  | Diploma | 42 | 13.50% |
|  | Bachelor | 126 | 40.40% |
|  | Masters | 15 | 4.80% |
|  | PhD | 11 | 3.50% |
|  | Professional Certificate | 56 | 17.90% |
| Marital Status | Married | 214 | 68.60% |
|  | Single | 98 | 31.40% |
| Firm Type | Private Sector | 212 | 67.9% |
|  | Public Sector | 100 | 32.10% |
| Rating of General Healthcare | Very Poor | 35 | 11.20% |
|  | Poor | 92 | 29.50% |
|  | Moderate | 116 | 37.20% |
|  | Good | 39 | 12.50% |
|  | Very Good | 30 | 9.60% |
| Experience with Drone Delivery | Yes | 72 | 23.10% |
|  | No | 240 | 76.90% |
| Opinion of Drone Policy | Continued | 224 | 71.80% |
|  | Suspended | 29 | 9. 30% |
|  | Indifferent | 59 | 18.90% |

Source: (Authors construct, 2022)

drone delivery services, majority of the staff have not directly experienced drone deliveries. This may be because, the health facilities have trained staff who pickup drone deliveries. The full demographic data is presented in Table 1.

## 4.2. Assessment of measures

Widely, the variance and covariance-based structural equation modelling techniques are used for model estimation and assessment. In this study, the variance-based approach (VB-SEM) was adopted for the model estimation and assessment leveraging SmartPLS—a partial least square structural equation modelling (PLS-SEM) software. Studies have shown that PLS-SEM has a high statistical power [71,72]. This implies that PLS-SEM is more likely to identify relationships as significant when they are indeed present in the population. This justifies the authors choice of variance-based approach with SmartPLS. Internal consistency tests (Cronbach's alpha and Composite Reliability), convergent validity tests (Average Variance Extracted), and discriminant validity tests (Fornell and Larcker criterion and Cross loading and Heterotrait-Monotrait Ratio) were adopted to assess the validity of the constructs. A summary of the assessment results is presented in Table 2.

**Table 2. Measures assessment results.**

| Details of measures, sources, and results of reliability tests for multi-item constructs | Factor Loadings |
|---|---|
| **Technology perceived usefulness** [92]: α = 0. 848; CR = 0. 898; AVE = 0. 688 <br> 1. Using drones in healthcare delivery improves the performance of health workers (PerUse1). <br> 2. Using drone healthcare delivery improves the efficiency of health delivery (PerUse2). <br> 3. Using drone healthcare delivery enhances the effectiveness in healthcare delivery (PerUse3). <br> 4. I find drone healthcare delivery to be useful in modern health delivery (PerUse4). | 0.817 <br> 0.790 <br> 0.832 <br> 0.877 |
| **Technology perceived benefits** [92]: α = 0. 901; CR = 0. 931; AVE = 0. 772 <br> 1. The use of drones in healthcare delivery helped save lives in the less privileged areas (PerBen1). <br> 2. Drone in healthcare delivery helped to reduce the mortality rates in the country (PerBen2). <br> 3. Drone in healthcare delivery improved the overall healthcare delivery in rural areas (PerBen3). <br> 4. Drone in healthcare delivery has brought significant benefits to our health institutions by improving outputs (PerBen4). | 0.881 <br> 0.892 <br> 0.858 <br> 0.882 |
| **Technology complexity** [93]: α = 0. 689; CR = 0. 865; AVE = 0. 762 <br> 1. The drones usage depends on sophisticated integration of technology, thereby making it complex to us in health delivery (CoPlex1) <br> 2. The drones usage is not complex because it provides flexible usage instruments you easily adjust to new demands (CoPlex2) <br> 3. The drone in healthcare delivery is very easy to access by the health institutions (CoPlex3) <br> 4. The technical characteristics of drones services, makes drone in healthcare delivery very complex (CoPlex4) | 0.512 <br> 0.850 <br> 0.895 <br> 0.542 |
| **Technology compatibility** [94]: α = 0. 815; CR = 0. 915; AVE = 0. 843 <br> 1. Drones technology is very compatible with the existing technologies at the health institutions (ComBTy1). <br> 2. The communication between service provided and end users is enhanced by the compatibility of the technologies being used (ComBTy2). <br> 3. The weather conditions makes drones in healthcare delivery operations very difficult (ComBTy3). <br> 4. There is lack of compatibility in using drones in healthcare delivery (ComBTy4). | 0.910 <br> 0.927 <br> 0.851 <br> 0.536 |
| **Technology perceived barrier** [95]: α = 0. 725; CR = 0. 832; AVE = 0. 558 <br> 1. There seems to be a lot of barriers in operations of the drone in healthcare delivery (PerBar1). <br> 2. My perception that the weather could be a barrier was right (PerBar2). <br> 3. My perception that the weather could be a barrier was wrong (PerBar3). <br> 4. There are a lot of financial barriers perceived to be affecting the operations of drones in healthcare delivery (PerBar4). | 0.856 <br> 0.728 <br> 0.562 <br> 0.809 |
| **Confirmation user expectation** [54]: α = 0. 788; CR = 0. 904; AVE = 0. 825 <br> 1. My usage experience of drones in healthcare delivery was better than expected (USexp1). <br> 2. The service level provided though drone in healthcare delivery operations was far better than expected (USexp2). <br> 3. Overall, most of expectation from drone in healthcare delivery usage experiment were confirmed (USexp3) <br> 4. I over expected the drone in healthcare delivery usage but my expectations were higher than what I personally experienced (USexp4). | 0.865 <br> 0.784 <br> 0.911 <br> 0.906 |
| **Confirmation perceived ease of use** [96]: α = 0. 848; CR = 0. 898; AVE = 0. 688 <br> 1. My intentions on drones in healthcare delivery is clear and understandable (PUse1). <br> 2. Using drones in healthcare delivery improves the efficiency of health delivery (PUse2). <br> 3. Using drones in healthcare delivery enhances the effectiveness in healthcare delivery (PUse3). <br> 4. I find drone healthcare delivery to be useful in modern health delivery (PUse4). | 0.839 <br> 0.889 <br> 0.906 <br> 0.879 |
| **User satisfaction prior usage** [97]: α = 0. 893; CR = 0. 926; AVE = 0. 757 <br> 1. My overall experience with the use of drones in healthcare delivery was very satisfying (PRioUse1). <br> 2. My overall experience with the drone in healthcare delivery was very pleased (PRioUse2). <br> 3. My overall experience with the drone in healthcare delivery was very contended (PRioUse3). <br> 4. My overall experience with the drone in healthcare delivery was absolutely delightful (PRioUse4). | 0.895 <br> 0.901 <br> 0.857 <br> 0.826 |

(*Continued*)

**Table 2.** (Continued)

| Details of measures, sources, and results of reliability tests for multi-item constructs | Factor Loadings |
|---|---|
| **User satisfaction IS expectations** [54]: α = 0. 834; CR = 0. 900; AVE = 0. 750 | 0.880 |
| 1. My expectation of using drones in healthcare delivery was met (ISExp1). | 0.886 |
| 2. My expectations of drones in healthcare delivery were really satisfied (ISExp2). | 0.733 |
| 3. My expectation of drones in healthcare delivery was dissatisfied (ISExp3). | 0.830 |
| 4. The initial expectation on the use of drone in healthcare delivery was met (ISExp4). | |
| **Environment government regulations** [56]: α = 0. 725; CR = 0. 845; AVE = 0. 645 | 0.803 |
| 1. The existing drones Laws in Ghana enhance the operations of drones in healthcare delivery (GovReg1). | 0.798 |
| | 0.808 |
| 2. Aviation laws do not allow drones to fly to a certain distance in Ghana (GovReg2). | 0.855 |
| 3. The service providers need the authorization from Aviation before a requested delivery is honored (GovReg3) | |
| 4. The Laws and regulations on drones are very wide than most advanced countries thereby make drone in healthcare delivery operation easy and prompt (GovReg4) | |
| **Environment trade partners** [90]: α = 0. 852; CR = 0. 900; AVE = 0. 692 | 0.803 |
| 1. There is over reliance on the trading partners of the drone in healthcare delivery operations (TPR1). | 0.839 |
| | 0.823 |
| 2. It seems the right partners were chosen for this drone in healthcare delivery (TPR2). | 0.862 |
| 3. Both the local and foreign partners are working hand in hand in the operations of drone in healthcare delivery (TPR3) | |
| 4. The partnership agreement is very open and transparent (TPR4). | |
| **Environment access to resources** [89]: α = 0. 838; CR = 0. 903; AVE = 0. 755 | 0.865 |
| 1. I am happy with the resources available for the operations of drones in healthcare delivery (ARS1). | 0.890 |
| | 0.852 |
| 2. Government has placed a high priority on resourcing the service providers (ARS2). | 0.893 |
| 3. All items listed as being part of healthcare delivery are promptly available in stock (ARS3). | |
| 4. There is always the shortage of drugs when there is an order (ARS4). | |
| **Organisation just-in-time delivery** [98]: α = 0. 837; CR = 0. 891; AVE = 0. 671 | 0.864 |
| 1. The order and delivery process is the best (JusTD1). | 0.798 |
| 2. There is 24 hours working time for orders and delivery of healthcare items (JusTD2). | 0.778 |
| 3. There seems to be the absence of delay in the delivery once an order has been made (JusTD3). | 0.833 |
| 4. Am satisfied with the speed of processing orders for delivery (JusTD4). | |
| **Organisation creativity** [99]: α = 0. 851; CR = 0. 910; AVE = 0. 771 | 0.899 |
| 1. The organization working with drone in healthcare delivery shows value for creativity (CretVTy1) | 0.886 |
| | 0.849 |
| 2. The value of organizational creativity is not made explicit to stakeholders (CretVTy2) | 0.902 |
| 3. The creativity of the organizations using drone in healthcare delivery makes its operation easier (CretVTy3) | |
| 4. The drone in healthcare delivery operating organizations has shown a great belief in creativity (CretVTy4) | |
| **Organisation efficiency** [100]: α = 0. 868; CR = 0. 919; AVE = 0. 792 | 0.902 |
| 1. I feel confident with the efficient operations of drone in healthcare delivery (EfFCy1) | 0.889 |
| 2. Service provider is efficiently working hand in hand with end users (EfFCy2) | 0.878 |
| 3. Drone in healthcare delivery is working efficiently (EfFCy3) | 0.803 |
| 4. Drone in healthcare delivery operations lacks efficiency (EfFCy4) | |
| **Organisation innovation** [101]: α = 0. 820; CR = 0. 893; AVE = 0. 735 | 0.855 |
| 1. Drone in healthcare delivery is very Innovative (INov1). | 0.857 |
| 2. Drone in healthcare delivery has shown an innovative way of healthcare delivery (INov2). | 0.859 |
| 3. The operation of drones in healthcare delivery lacks innovation (INov3). | 0.768 |
| 4. Using drones in healthcare delivery shows a lot of innovations being added to healthcare delivery (INov4). | |
| **Organisation leadership** [33]: α = 0. 750; CR = 0. 842; AVE = 0. 576 | 0.841 |
| 1. The drone in healthcare delivery operators has shown leadership in delivery (Leaship1). | 0.821 |
| 2. Good leadership will bring about a lot of innovations into healthcare delivery (Leaship2). | 0.566 |
| 3. There is a lack of leadership in the operations of drones in healthcare delivery (Leaship3). | 0.776 |
| 4. I have seen true leadership in the operations of drones in healthcare delivery (Leaship4). | |

**NB:** α = Cronbach's alpha; CR = Composite reliability; AVE = Average variance extracted

Source: authors construct, 2022.

Convergent validity test was carried out to measure the degree to which each measure correlates favourably with an alternate measure of the same construct. The test was carried out by accessing the resulting values of the Average Variance Extracted (AVE). The results showed that all the measures had their AVE statistics higher than the recommended threshold of .50 [73] indicating that all the measures are adequately valid.

Further, the discriminant validity test was carried out to examine the extent to which each construct is truly distinct from the other constructs. Fornell and Larcker criterion, cross loading and Heterotrait-Monotrait Ratio (HTMT) were used to assess the discriminant validity. According to [73] criterion, discriminant validity is ascertained when the square root of AVE for a construct is greater than its correlation with all other constructs. It was shown that the square root or leading diagonal elements were all greater than the corresponding constructs estimates (correlation coefficient of each latent variable), providing a strong support for the establishment of discriminant validity. Also, all the item factor loadings are heavier on the underlying construct to which they belong rather than the different construct in the study [74], providing further evidence for discriminant validity using the cross-loading statistics. Heterotrait-Monotrait Ratio (HTMT) for the constructs were also less than the acceptable threshold of 0.90 [75]. Finally, an internal consistency test was carried out to examine how the collection of indicators accurately measured the latent variables. Cronbach's alpha and composite reliability statistics were used for the investigation. The examination of the data revealed that all the constructs had a Cronbach alpha higher than the recommended threshold, 0.7 [72] except technology complexity, (that reported alpha value 0.689). However, all the constructs including technology complexity had composite reliability test values exceeding the recommended threshold 0.7 [72]. Hence, construct reliability is established.

## 4.3 Findings

The previous section established the validity of constructs indicating that they are reliable enough to generate accurate insights for the study. This section examines the relationship between the validated constructs following the proposed hypothesis. The results of the structural model are presented in Table 3 below:

An assessment was carried out to validate the proposed hypothesis (H1a = There is a positive relationship between technology adoption and user satisfaction; H1b = There is a positive relationship between confirmation of expectation and user satisfaction; H2 = There is a direct relationship between user satisfaction and actual usage. H3 = There is a positive relationship between actual usage and continuance usage; H4 = There is a direct relationship between

**Table 3. Results of structural model.**

| Path | β | P-Value | Decision |
|---|---|---|---|
| H1a: Technology -> User Satisfaction | 0.425 | 0.000 | Supported |
| H1b: Confirmation -> User Satisfaction | 0.288 | 0.000 | Supported |
| H2: User Satisfaction -> Actual Usage | 0.412 | 0.000 | Supported |
| H3: Actual Usage -> Continuous Usage | 0.340 | 0.000 | Supported |
| H4: Organization -> Continuous Usage | 0.507 | 0.000 | Supported |
| H5a: Environment->User Satisfaction | 0.174 | 0.018 | Supported |
| H5b: Environment->Actual Usage | 0.412 | 0.000 | Supported |
| H5c: Environment->Continuance Usage | -0.033 | 0.731 | Not supported |
| H6: Environment x Actual Usage -> Continuance Usage | -0.054 | 0.020 | Supported |

Source: (Authors construct, 2022)

organization and continuance usage; H5a = There is a positive relationship between environment and user satisfaction; H5b = There is a positive relationship between environment and actual usage; H5c = There is a positive relationship between environment and continuous usage; and H6 = Environment moderates the relationship between actual usage and continuous usage).

The findings of the assessment as shown in Table 3 above revealed that there is a positive relationship between technology adoption and user satisfaction (H1a: $\beta$ = 0.425, p = 0.000, supported). Confirmation of expectation was found to positively relate with user satisfaction (H1b: $\beta$ = 0.288, p = 0.000, supported). User satisfaction was found to positively affect actual usage (H2: $\beta$ = 0.412, p = 0.000, supported) which eventually leads to continuous usage (H3: $\beta$ = 0.340, p = 0.000, supported). Organisational variables like just-in-time delivery, creativity, efficiency, innovations, and leadership were found to have a direct relationship with information systems continuance usage (H4: $\beta$ = 0.507, p = 0.000, supported). It was also revealed that there is a direct relationship between the organisation's operational environment (government regulation, trading partners and, access to resources) and user satisfaction (H5a: $\beta$ = 0.174, p = 0.018, supported); and actual usage (H5b: $\beta$ = 0.412, p = 0.000, supported). However, the hypothesis for the relationship between the environment and continous usage was not supported (H5c: $\beta$ = -0.033, p = 0.731, not supported). Finally, the relationship between actual usage and continuous usage was found to be moderated by the environment (H6: $\beta$ = -0.054, p = 0.020, supported).

## 5. Discussion and implications

This study examined the factors accounting for drone adoption and continuance usage in healthcare delivery. Specifically, the study focused on the role of the environment in drone adoption and continuance usage in healthcare delivery. To empirically investigate the issue, data were collected from health facilities that engage in the use of Zipline's drone services as well as Zipline Ghana. The data collected was then modelled to examine the factors accounting for and the role of the environment in drone adoption and continuance usage for healthcare delivery. The results of the analysis of the data from a developing economy showed that there is a positive relationship between technology adoption and user satisfaction. Again, there is a positive relationship between confirmation of expectation and user satisfaction. In addition, user satisfaction positively affects actual usage which eventually leads to continuous usage. Moreover, there was a direct relationship between organization and information systems continuance usage. It was also shown that there is a direct relationship between the environment and user satisfaction, and actual usage. Finally, the environment moderated the relationship between actual usage and continuous usage. These findings contribute to the theoretical debates in the drone and healthcare delivery literature.

### 5.1. Relationship between technology, confirmation and user satisfaction

The study showed that there is a significant and positive relationship between technology and user satisfaction (H1a). Again, there is a positive relationship between confirmation and user satisfaction (H1b) which showed consistency with empirical literature and provided further confirmation to the research theories. First, studies have revealed a positive relationship between users' confirmation of their perceived usefulness and satisfaction [76,77]. This implies that if users believe that an IS is very useful and the actual use experience corresponds or goes beyond their initial expectation, the confirmation that exists leads to user satisfaction [78,79]. This is because the expected benefits of IS use are realized [80]. With regard to technology, studies have highlighted that the complexity of smart services is related to users' discomfort

with smart technologies. Thus, when users' feeling of discomfort with technology is high, it is likely to have a negative impact on their ability to use smart services, and as such, will have negative attitudes towards its adoption [81]. As a result, the success of smart services should not focus only on the quality of the actual service delivered to the users but also on the complexity or simplicity of the software that drives the smart systems interface hardware. This may also be extended in the adoption of drone technology usage and user satisfaction. This implies that a high level of functionality of drone technology will result in improved levels of user satisfaction. In order to eliminate or, minimize the complexity of smart service delivery channels, the interface technologies should be efficient and personalize it in order to give the users sufficient choices and information to make real-time decisions and accomplish their service needs easily and timely without being confounded by the smart service delivery technologies [82]. In relation to the confirmation of expectations, it is key that smart service delivery providers like drone delivery technologies involve and interact with users through the whole cycle of service development to minimize user challenges [48]. Expectation before initial usage provides a baseline for users to evaluate their satisfaction. Users' confirmation of expectations suggests that the users obtained expected value through their usage experiences with the IS and thus leads to a positive effect on users' satisfaction with their initial IS use [55].

## 5.2. Relationship between user satisfaction and actual usage

The study showed that there is a positive relationship between user satisfaction and actual usage (H2) as hypothesized. This finding is consistent with empirical literature and provided further confirmation to the research theories. [83] discusses a lower bound to satisfaction below which users may discontinue system use, which gives some direct insight into how user satisfaction influences system usage. Using path analysis, [84] argued that user information satisfaction (UIS) could lead to system usage. In general, perceived user-friendliness or ease-of-use of technological innovations offers the strongest explanation of the impacts of such technologies on users' satisfaction and behavioral intention to use such innovations [48]. According to the expectation-confirmation theory, consumer's satisfaction refers to their cognitive and affective state of fulfillment after a purchase. It is a key determinant of repurchase intention [55]. As such, by extending the interpretations to the findings of this study, by achieving higher levels of user satisfaction, there will be an enhancement in the actual usage and drone technology for the delivery of health services in Ghana.

## 5.3. Relationship between actual usage and continuance usage

The study revealed that there is a positive relationship between actual usage and continuance usage (H3) which is consistent with empirical literature and provided further confirmation to the research theories. Users' experience of technology application is a major criterion for the evaluation of customer's satisfaction with the services [85]. As such, improved technological innovation and functionality increase user satisfaction. Studies have shown that user information satisfaction (UIS) could lead to system usage [84]. This argument has been extended by [86] and posits that satisfaction is an intrinsic motivator in using information systems. Thus, the actual usage of systems is promoted. Not much had, however, been done to support the position of actual usage supporting continuance usage. We argue from the factors which promote actual usage and explain that through an actual interaction and experience with technology, users may directly experience its benefits, confirm its usefulness and further clear any perceptions about barriers and constraints leading to the continuance usage of the technology.

### 5.4. Relationship between organization and continuance usage

The study revealed that there is a significant and positive relationship between organization and continuance usage (H4). This indicates that the organizational variables just-in-time delivery, creativity, efficiency, innovations, and leadership [51] influences the continuance usage of drone technologies in healthcare delivery. In terms of organizational issues, [87] opine that critical consideration be given when implementing large and complex information systems and technology, particularly in regard to top management support and facilitating conditions for the information systems and technology development within medical environments. Extending the findings in the adoption of drone technology, organizational factors will rather determine the continuance usage of drone technology.

### 5.5. Relationship between environment, user satisfaction, actual and continuance usage

The study showed that there is a significant and positive relationship between environment and user satisfaction (H5a). In addition, there is a significant and positive relationship between environment and actual usage (H5b) however, the relationship between environment and continuance usage (H5c) was not significant. As suggested by [88], environmental issues represent the current operating environment of the healthcare industry, and it is worth mentioning that government policies should be particularly considered when implementing the new information systems and technologies within medical environments. In the context of environmental factors, particularly in government regulations, [88] confirms that government policy has a significant impact on hospitals' information systems adoption decision. If the government keeps its promise on developing the information systems, this will cause users to have a positive feeling that the IS/IT will be useful. However, the continuance usage may not necessarily be impacted by these factors. It can therefore be said that environmental factors contribute to actual usage of drone technology service. External environmental factors like government policy, trade partners, access to resources may have already been contributory to the actual usage. Thus, a zero or reduced levels of such factors will probably lead to continuance usage, as other internal factors of the service will be more significant towards continuance usage.

### 5.6. Moderating role of environment in the relationship between actual and continuance usage

It was revealed that the environment moderates the relationship between actual usage and continuance usage (H6). From the results, this relationship was found to be significant indicating that the relationship between actual and continual usage of drone services in healthcare delivery is contingent upon environmental factors such as access to resources [86,89,90]. However, this interaction was weak and negative suggesting that reducing the influence of environmental factors will rather increase the impact actual usage have on the continuance usage of drone technology in the delivery of healthcare services. Since no such study is attributed to this finding, no connection can be made with previous studies. Nonetheless, since earlier findings confirmed the limited interaction of environmental factors and continuance usage, the weakness of the moderating effect can provide further confirmation to the limited effect environmental factors have on the continuance usage of drone technology services in the healthcare delivery services.

## 6. Summary and conclusion

This study investigated and contributed to the debate on drone technology usage and the interaction of technology, confirmation, user satisfaction, actual usage, organisation, environment

and continuous usage. It is evident that technological and confirmation factors contribute to the attitude of adopting drone technologies in healthcare delivery. Findings from the demographics indicated that, a small number of respondents 72(23.10%) indicated their experience with the use of drone technology services in health care delivery in Ghana showing a relatively low level of its adoption currently. However, most respondents 224(71.80%) advocated for the continued use of drone technology in health care policy, indicating the willingness of stakeholders to experience the service, its perceived issues and potential benefits. The actual usage of drone technology services is impacted by user satisfaction factors (prior usage and IS expectations) and environmental factors (government, access to resources and trade partners). With regard to user satisfaction factors, the technology factors and confirmation factors ought to be significant to contribute to satisfying user needs in order for them to pursue an actual usage of the technology. The availability of government regulation and policy, resources and trade partners also facilitate drone use in the health sector. It was found that users are more likely to keep using drone technology after having actual interaction with the service. The environment (government, access to resources and trade partners) contributed to drone technology deployment as it was significant to user satisfaction and actual usage. Thus, the availability of high levels of these factors would likely form part of initial drone technology adoption in healthcare delivery. Organisation (efficiency, creativity, innovation, leadership and just in time delivery) was found to be significant towards continuance usage of drone technology services. This implies that, upon increasing the levels of these factors by the services providers, users are more likely to continue using the service.

Following the findings of the study, confirmation and technology factors are influential in achieving user satisfaction, which in turn influences the actual usage of drone technology services. As such, it is important that issues like improved functionality, benefits, compatibility and reduced complexity are keenly considered in the provision of drone services since these are likely to influence the levels of user satisfaction, usage and subsequently continuance usage. Service providers should implement strategies to receive feedback from users served with the service. Their satisfaction levels can be ascertained, and further measures may be undertaken to continue its usage. Service providers should also consider improving the levels of efficiency, creativity, innovation and delivery of the service to ensure continuance usage of drone technologies in healthcare delivery. In addition, government policy on drone technology should further be strengthened to improve levels of actual usage in different areas across the country. The policy should consider making provision for adequate and sufficient number of resources, both financial and technical, to facilitate the deployment of drone technologies in healthcare delivery. Specific trade partners may also be considered and partnered with to promote technology transfer and knowledge sharing on the use of the systems, which may all be critical to its actual usage.

The study was scoped to Zipline drone delivery services in the Ghana Health Service. In addition, this research was based on data collected from a self-report questionnaire made up of multiple questions. The results of the data collected may have been limited due to response bias [91]. This makes the findings less generalizable to the use of drone technology in other sectors. It is suggested that future studies explore the use of drone technology deliveries within other sectors like hospitality, agriculture and e-commerce. Future studies should explore different interactions that exist amongst the variables and how they contribute to drone technology adoption and continuance usage.

## Supporting information

**S1 Data. This file contains the output data from our PLS analysis of drone data.**
(XLSX)

**S2 Data. This file includes the bootstrap analysis results for our drone study.**
(XLSX)

**S1 Fig. This image displays the structural equation model (SEM) related to our drone research.**
(PNG)

## Author Contributions

**Conceptualization:** John Serbe Marfo.

**Data curation:** Kwadwo Kyeremeh.

**Methodology:** Kwadwo Kyeremeh.

**Project administration:** John Serbe Marfo.

**Writing – original draft:** Pasty Asamoah.

**Writing – review & editing:** John Serbe Marfo, Matilda Kokui Owusu-Bio, Afia Frimpomaa Asare Marfo.

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
