## [Decision Letter · Decision Letter 0]

5 Jul 2023

PDIG-D-23-00170

Exploring Factors Affecting the Adoption and Continuance Usage of Drone in Healthcare: The Role of the Environment

PLOS Digital Health

Dear Dr. Marfo,

Thank you for submitting your manuscript to PLOS Digital Health. After careful consideration, we feel that it has merit but does not fully meet PLOS Digital Health's publication criteria as it currently stands. Therefore, we invite you to submit a revised version of the manuscript that addresses the points raised during the review process.

Please submit your revised manuscript within 60 days Sep 03 2023 11:59PM. If you will need more time than this to complete your revisions, please reply to this message or contact the journal office at digitalhealth@plos.org. Please include the following items when submitting your revised manuscript:

We look forward to receiving your revised manuscript.

Kind regards,

Laura M. König

Academic Editor

PLOS Digital Health

Journal Requirements:

1. We ask that a manuscript source file is provided at Revision. Please upload your manuscript file as a .doc, .docx, .rtf or .tex.

2. Please provide separate figure files in .tif or .eps format only and remove any figures embedded in your manuscript file. Please also ensure that all files are under our size limit of 10MB.

Additional Editor Comments (if provided):

Reviewers' comments:

Reviewer's Responses to Questions

**Comments to the Author**

1. Does this manuscript meet PLOS Digital Health’s publication criteria? Is the manuscript technically sound, and do the data support the conclusions? The manuscript must describe methodologically and ethically rigorous research with conclusions that are appropriately drawn based on the data presented.

Reviewer #1: Partly

Reviewer #2: Yes

2. Has the statistical analysis been performed appropriately and rigorously?

Reviewer #1: I don't know

Reviewer #2: Yes

3. Have the authors made all data underlying the findings in their manuscript fully available (please refer to the Data Availability Statement at the start of the manuscript PDF file)?

Reviewer #1: No

Reviewer #2: Yes

4. Is the manuscript presented in an intelligible fashion and written in standard English?

Reviewer #1: Yes

Reviewer #2: Yes

5. Review Comments to the Author

Reviewer #1: This paper seeks to explore the role of environment in the usage and continued usage of drones to deliver healthcare supplies in Ghana. This provides an interesting overview, but there are some problems with the current manuscript. In particular the authors posit that this is a survey of drone users, At the beginning of the section 3.2.2, it states that “The participants of the study included only staff who make use of drugs and health commodities delivered through drones but 76% of the respondents had not experienced drone delivery according to the demographics. The authors should explain this and how it impacts their analysis. 

Section 2.4

• Need references to theories being used in the theoretical framework

• The ECT is a theory to explain purchase and use, but who are the users in this context – is this a survey of the health institutes or the patients receiving the care?

• "ECT has a limitation in explaining the information system expectation formation process" (Yang, 2017). – what is this limitation and how is it relevant

Questions table

Some of the phrasing of questions is problematic as there are multiple decisions to be made within single question, this makes it difficult to understand the exact nature of influence. . 

It would be useful to see the full factor mappings table, as I am sure some items also map on to other constructs. The way decisions were made as to which factor they belonged to should be explained. 

The construct of organisation creativity needs more explanation as to its relevance to this work, 

Unclear how the individual factors are used to calculate the constructs, eg environment construct is made up of multiple factors, how do they relate to each other, and why this particular focus? 

Results

The paper states that Convergent validity is measured against alternative measures – but what are these measures?

Reviewer #2: This is a nice body of evidence which will add to the emerging tech space . Authors elaborated a lot in explaining the concepts and methodology, suggest to elaborate more on results and bring in more substance into the discussion. Make the literature search and concept description short

6. PLOS authors have the option to publish the peer review history of their article (what does this mean?). If published, this will include your full peer review and any attached files.

**Do you want your identity to be public for this peer review?** For information about this choice, including consent withdrawal, please see our Privacy Policy.

Reviewer #1: No

Reviewer #2: Yes: Shibu Vijayan

---

## [Decision Letter · Decision Letter 1]

13 Sep 2023

Exploring Factors Affecting the Adoption and Continuance Usage of Drone in Healthcare: The Role of the Environment

PDIG-D-23-00170R1

Dear Dr Marfo,

We are pleased to inform you that your manuscript 'Exploring Factors Affecting the Adoption and Continuance Usage of Drone in Healthcare: The Role of the Environment' has been provisionally accepted for publication in PLOS Digital Health.

Best regards,

Laura M. König

Academic Editor

PLOS Digital Health

Reviewer Comments (if any, and for reference):

Reviewer's Responses to Questions

**Comments to the Author**

1. If the authors have adequately addressed your comments raised in a previous round of review and you feel that this manuscript is now acceptable for publication, you may indicate that here to bypass the “Comments to the Author” section, enter your conflict of interest statement in the “Confidential to Editor” section, and submit your "Accept" recommendation.

Reviewer #2: All comments have been addressed

2. Does this manuscript meet PLOS Digital Health’s publication criteria? Is the manuscript technically sound, and do the data support the conclusions? The manuscript must describe methodologically and ethically rigorous research with conclusions that are appropriately drawn based on the data presented.

Reviewer #2: (No Response)

3. Has the statistical analysis been performed appropriately and rigorously?

Reviewer #2: I don't know

4. Have the authors made all data underlying the findings in their manuscript fully available (please refer to the Data Availability Statement at the start of the manuscript PDF file)?

Reviewer #2: No

5. Is the manuscript presented in an intelligible fashion and written in standard English?

Reviewer #2: Yes

6. Review Comments to the Author

Reviewer #2: (No Response)

7. PLOS authors have the option to publish the peer review history of their article (what does this mean?). If published, this will include your full peer review and any attached files.

**Do you want your identity to be public for this peer review?** For information about this choice, including consent withdrawal, please see our Privacy Policy.

Reviewer #2: **Yes: **Shibu Vijayan
